# NEURAL RADIANCE FIELD CODEBOOKS

**Matthew Wallingford**[1], **Aditya Kusupati**[1], **Alex Fang**[1], **Vivek Ramanujan**[1],
**Aniruddha Kembhavi**[2], **Roozbeh Mottaghi**[1], **Ali Farhadi**[1]
[1]University of Washington; [2]PRIOR, Allen Institute for AI

## ABSTRACT

Compositional representations of the world are a promising step towards enabling high-level scene understanding and efficient transfer to downstream tasks. Learning such representations for complex scenes and tasks remains an open challenge. Towards this goal, we introduce Neural Radiance Field Codebooks (NRC), a scalable method for learning object-centric representations through novel view reconstruction. NRC learns to reconstruct scenes from novel views using a dictionary of object codes which are decoded through a volumetric renderer. This enables the discovery of reoccurring visual and geometric patterns across scenes which are transferable to downstream tasks. We show that NRC representations transfer well to object navigation in THOR, outperforming 2D and 3D representation learning methods by 3.1% success rate. We demonstrate that our approach is able to perform unsupervised segmentation for more complex synthetic (THOR) and real scenes (NYU Depth) better than prior methods (29% relative improvement). Finally, we show that NRC improves on the task of depth ordering by 5.5% accuracy in THOR.

## 1 INTRODUCTION

Parsing the world at the abstraction of objects is a key characteristic of human perception and reasoning (Rosch et al., 1976; Johnson et al., 2003). Such object-centric representations enable us to infer attributes such as geometry, affordances, and physical properties of objects solely from perception (Spelke, 1990). For example, upon perceiving a cup for the first time one can easily infer how to grasp it, know that it is designed for holding liquid, and estimate the force needed to lift it. Learning such models of the world without explicit supervision remains an open challenge.

Unsupervised decomposition of the visual world into objects has been a long-standing challenge (Shi & Malik, 2000). More recent work focuses on reconstructing images from sparse encodings as an objective for learning object-centric representations (Burgess et al., 2019; Greff et al., 2019; Locatello et al., 2020; Lin et al., 2020; Monnier et al., 2021; Smirnov et al., 2021). The intuition is that object encodings which map closely to the underlying structure of the data should provide the most accurate reconstruction given a limited encoding size. Such methods have shown to be effective at decomposing 2D games and simple synthetic scenes into their parts. However, they rely solely on color cues and do not scale to more complex datasets (Karazija et al., 2021; Papa et al., 2022).

Advances in neural rendering (Mildenhall et al., 2021; Yang et al., 2021) have enabled learning geometric representations of objects from 2D images. Recent work has leveraged scene reconstruction from different views as a source of supervision for learning object-centric representations (Stelzner et al., 2021; Yu et al., 2021b; Sajjadi et al., 2022a; Smith et al., 2022). However, such methods have a few key limitations. The computational cost of rendering scenes grows linearly with the number of objects which inhibits scaling to more complex datasets. Additionally, the number of objects per scene is fixed and fails to consider variable scene complexity. Finally, objects are decomposed on a per scene basis, therefore semantic and geometric information is not shared across object categories.

With this in consideration we introduce, Neural Radiance Codebooks (NRC). NRC learns a codebook of object categories which are composed to explain the appearance of 3D scenes from multiple views. By reconstructing scenes from different views NRC captures reoccurring geometric and visual patterns to form object categories. This learned representation can be used for segmentation as well as geometry-based tasks such as object navigation and depth ordering. Furthermore, NRC resolves the limitations of current 3D object-centric methods. First, NRC's method for assigning object

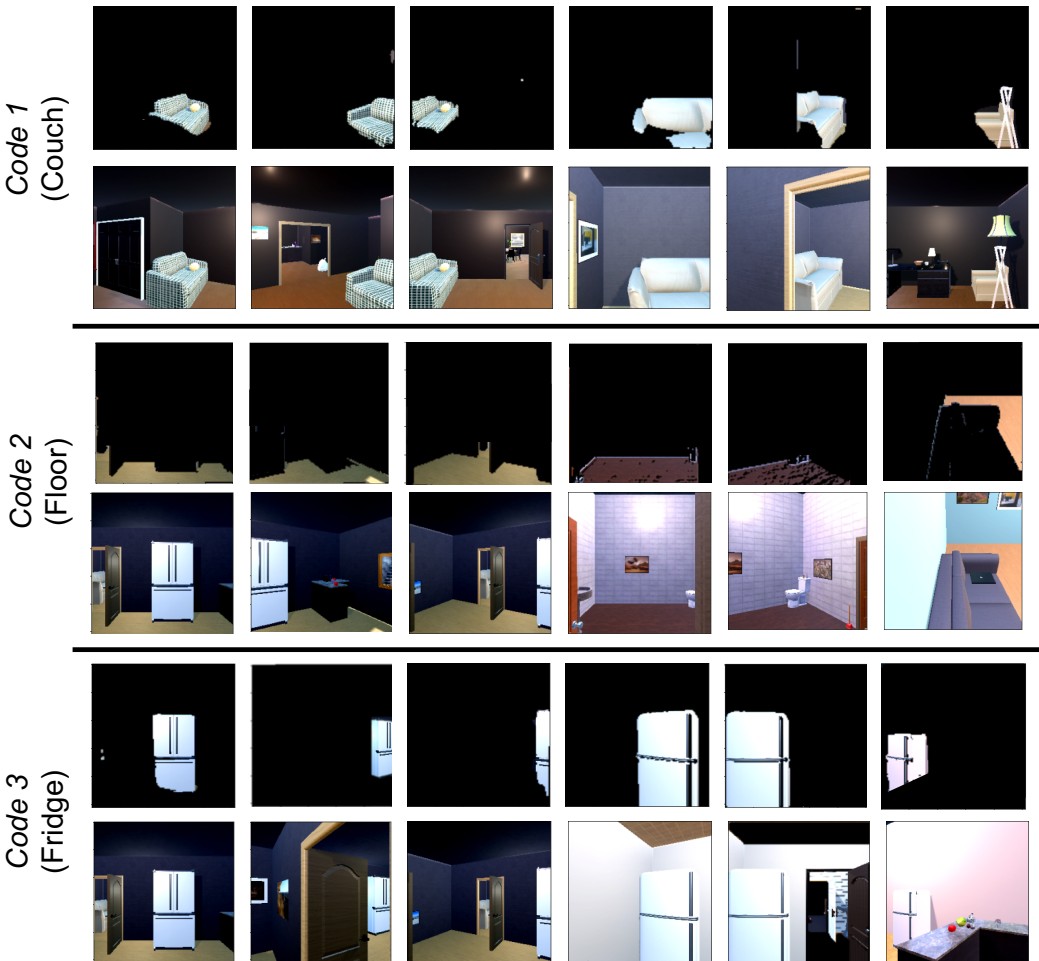

Figure 1: **Visualization of learned codes.** The NRC codebook encodes reoccurring geometric and visual patterns. In the top row, couches of differing appearance are grouped by geometric structure. In the middle row, different textured floors are categorized based on their shared planar geometry. In the bottom row, NRC learns correspondences between fridges from different views and scenes.

codes to regions of the image enables *constant rendering compute* whereas that of other methods scales with number of objects. Second, we introduce a novel mechanism for differentiably adding new categories which allows the codebook to scale with the complexity of the data. Last, modeling intra-category variation in conjunction with the codebook enables sharing of semantic and geometric object information across scenes.

We evaluate NRC on unsupervised segmentation, object navigation, and depth ordering. For segmentation on indoor scenes from ProcTHOR (Deitke et al., 2022) we show 29.4% relative ARI improvement compared to current 3D object-centric methods (Stelzner et al., 2021; Yu et al., 2021b). On real-world images (NYU Depth (Silberman et al., 2012)) we show promising qualitative results (Figure 3) and 29% relative improvement. For object navigation and depth ordering, where geometric understanding is relevant, we observe 3.1% improvement in navigation success rate 5.5% improvement in depth ordering accuracy over comparable self-supervised and object-centric methods. Interestingly, we find qualitative evidence that the learned codes categorize objects by both visual appearance and geometric structure (Figure 1).

## 2 RELATED WORK

**Object-Centric Learning**   Object-centric learning aims to build compositional models of the world from building blocks which share meaningful properties and regularities across scenes. Prior works

such as MONet (Burgess et al., 2019), IODINE (Greff et al., 2019), Slot Attention (Locatello et al., 2020), and Monnier et al. (2021) have demonstrated the potential for disentangling objects from images. Other work has shown the ability to decompose videos (Kabra et al., 2021; Kipf et al., 2021). In particular, Marionette (Smirnov et al., 2021) learns a shared dictionary for decomposing scenes of 2D sprites. We draw inspiration from MarioNette for learning codebooks, but differ in that we model the image formation process and intra-code variation, and dynamically add codes to our dictionary.

**3D Object-Centric Learning**   Recent work has shown novel view reconstruction to be a promising approach for disentangling object representations. uORF (Yu et al., 2021b) and ObSuRF (Stelzner et al., 2021) combine Slot Attention with Neural Radiance Fields (Mildenhall et al., 2021) to decompose scenes. COLF (Smith et al., 2022) replaces the volumetric renderer with light fields to improve computational efficiency. NeRF-SOS Fan et al. (2022) uses contrastive loss for both geometry and appearance to perform object segmentation. SRT (Sajjadi et al., 2022b) encodes scenes into a set of latent vectors which are used to condition a light field. OSRT (Sajjadi et al., 2022a) extends SRT by explicitly assigning regions of the image to latent vectors. NeSF Vora et al. (2021) learns to perform 3D object segmentation using NeRF with 2D supervision. Although great progress has been made, these methods are limited to synthetic and relatively simple scenes. Our work differs from previous 3D object-centric works in that we learn reoccurring object codes across scenes and explicitly localize the learned codes. Additionally, our method can model an unbounded number of objects per scene compared to prior work which fixes this hyper-parameter a priori. We show that our approach generalizes to more complex synthetic and real-world scenes.

**Neural Rendering**   Advances in neural rendering, in particular Neural Radiance Fields (NeRF) (Mildenhall et al., 2021), have enabled a host of new applications (Jang & Agapito, 2021; Mildenhall et al., 2022; Pumarola et al., 2021; Park et al., 2021; Lazova et al., 2022; Niemeyer & Geiger, 2021; Zhi et al., 2021). NeRF differentiably renders novel views of a scene by optimizing a continuous volumetric scene function given a sparse set of input views. Original formulation of NeRF learned one representation for each scene; other works (Yu et al., 2021a; Jain et al., 2021; Kosiorek et al., 2021) showed conditioning NeRFs on images enables generation of novel views of new scenes.

**Dictionary/Codebook Learning**   Dictionary (codebook) learning (Olshausen & Field, 1997) involves learning of a specific set of atoms or codes that potentially form a basis and span the input space through sparse combinations. Codebooks have been widely used for generative and discriminative tasks across vision (Elad & Aharon, 2006; Mairal et al., 2008), NLP (Mcauliffe & Blei, 2007) and signal processing (Huang & Aviyente, 2006). Learning sparse representations based on codes enables large-scale methods which rely on latent representations. More recently, codebooks have been shown to be crucial in scaling discrete representation learning (Van Den Oord et al., 2017; Kusupati et al., 2021). Marionette (Smirnov et al., 2021) is an object-centric representation learning method that relies on codebooks, unlike most other methods that are developed around set latent representations (Sajjadi et al., 2022a;b; Locatello et al., 2020; Yu et al., 2021b). Object-centric codebooks help in semantic grounding for transfer between category instances and are important for large-scale representation learning across diverse scenes and objects.

## 3   METHOD

Our goal is to discover object categories without supervision, learn priors over their geometry and visual appearance, and model the variation between instances belonging to each group. Given multiple views of a scene, the objective is to explain all views of the scene given a set of object-codes. This learned decomposition can be used for segmentation and other downstream tasks that require semantic and geometric understanding such as depth ordering and object-navigation.

Figure 2 illustrates the training pipeline. We begin by processing the image through a convolutional network to obtain a spatial feature map. The feature map is then projected to a novel view using the relative camera matrix. Feature vectors from each respective region of the image are assigned to categorical latent codes from the finite-size codebook. The object codes and feature vectors are passed to a convolutional network which transform the categorical codes to fine-grained instance codes. A volumetric renderer is then conditioned on the instance code, view direction, and positional encodings to render each region of the scene from the novel view. The rendered image from the

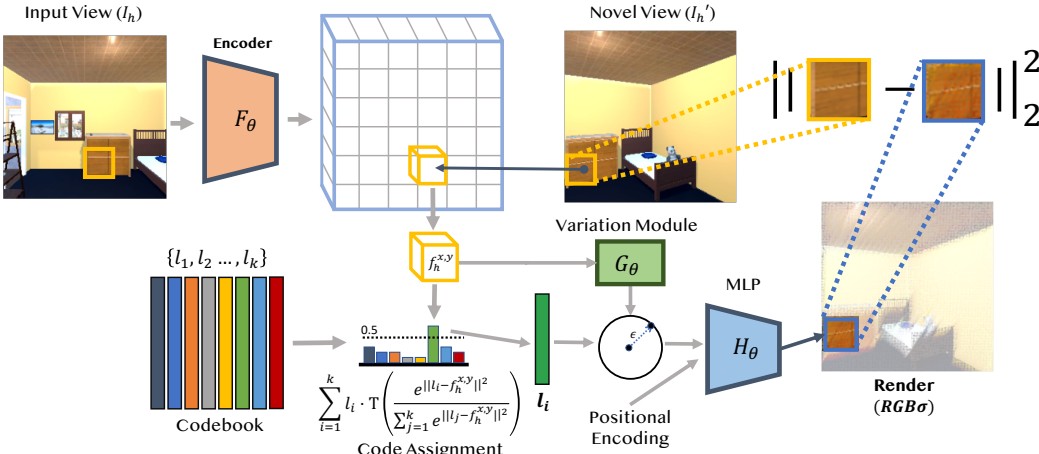

Figure 2: An overview of NRC. We learn a set of shared codes for decomposing scenes into objects. Each point in the scene is assigned one of $n$ latent codes from the codebook. The variation module models the intra-code variation between objects by perturbing the code in latent space. A conditional NERF model renders the scene and is compared to the ground truth novel view for supervision.

novel view is compared to the ground truth using $L_2$ pixel loss. The categorical codes, assignment mechanism, and volumetric renderer are learned jointly in an end-to-end fashion.

**Image Encoding and Camera Projection**  Given an input frame and novel image, $I_h, I_h' \in \mathcal{R}^{3 \times H \times W}$ respectively from scene $S_h$, we first encode $I_h$ into a spatial feature map, $f_h \in R^{d \times H/k \times W/k}$, using a convolutional network, $F_\theta$. We project each point, $(x, y, z)$, in world coordinates of the novel view to camera coordinates in the input frame, $(x, y)$, using the relative camera pose. Given $(x, y)$, we select the spatial feature $f_h^{x,y} \in \mathcal{R}^d$ from the patch that contains the projected coordinates. The spatial features vectors are then passed to the next stage where they are assigned to categorical object-codes.

**Assigning + Learning Codes**  Our goal is to jointly learn a shared set of object categories and priors about their appearance and geometry. By mapping the spatial features from a continuous vector space to a discrete, finite set of codes the model is incentivized to find reoccurring patterns in the images.

Given the features for a point in the novel view, $f_h^{x,y}$, we assign a code, $l^*$, chosen from the shared codebook, $\mathbb{L}$. We do so with an $\arg\max$ 1-nearest-neighbors during inference:

$$l^*(x, y) \leftarrow \underset{l_i;\ i\in[k]}{\overset{(\text{STE})}{\arg\max}} \frac{e^{-\|l_i - f_h^{x,y}\|_2}}{\sum_{j=1}^{k} e^{-\|l_j - f_h^{x,y}\|_2}} \tag{1}$$

The nearest-neighbor assignment used during inference is a non-differentiable operation therefore propagating gradients to the encoder and codebook would not be possible. To enable learning of the codebook elements we use a softmax relaxation of nearest-neighbors during the backward pass in conjunction with the straight-through-estimator (STE) Bengio et al. (2013):

$$l_{back}^*(x, y) \leftarrow \frac{e^{-\|l_i - f_h^{x,y}\|_2}}{\sum_{j=1}^{k} e^{-\|l_j - f_h^{x,y}\|_2}} \tag{2}$$

**Adding Categorical Codes**  The number of codes should depend on the complexity of the scenes they model. Learning when to add new codes is non-trivial because the number and selection of codes is discrete and non-differentiable. To circumvent this problem, we use a series of step functions with a straight-through-estimator (STE) to sequentially add elements to the codebook. Each code $l_i \in \mathbb{L}$ is gated according to the following:

$$l_i \leftarrow \mathcal{T}\left(\boldsymbol{\sigma}(s - i^2/\lambda), \frac{1}{2}\right) \cdot l_i; \quad \mathcal{T}(a, t) := \begin{cases} 1, & a > t \\ 0, & a \leq t \end{cases} \tag{3}$$

$\mathcal{T}(.)$ is a binarization function in the forward pass and lets the gradients pass through using STE in the back pass. $\boldsymbol{\sigma}(\cdot)$ is the sigmoid function, $\lambda$ is a scaling hyperparameter, and $s$ is a learnable scoring parameter whose magnitude is correlated with the overall capacity (number of codes) required to model the scenes accurately. A new code $l_i$ is added when $s$ exceeds the threshold $i^2/\lambda$. New codes are initialized using a standard normal distribution. Throughout training we keep $k+1$ total codes where $k$ is the current number of learnable codes. The extra code is used by the straight-through-estimator to optimize for $s$ on the backward pass. This formulation can be viewed as the discrete analog of a gaussian prior over the number of elements, $k$ in the codebook: $P(k) = e^{-k^2/\lambda}$.

**Modeling Intra-Code Variation**    Once a categorical code has been assigned to a region in the novel frame, the model must account for variation across instances. We model this variation in latent space using an encoder that takes in both the spatial feature, $f_h^{x,y}$, and the categorical code $l^*(x,y)$. We rescale the norm of the variation vector by $\epsilon$, a hyperparameter, to ensure the instance and categorical codes are close in latent space. The instance code is formulated as the following:

$$l^*_{\text{instance}}(x,y) = l^*(x,y) + \epsilon \cdot \frac{G_{\theta'}([l^*(x,y), f_h^{x,y}])}{||G_{\theta'}([l^*(x,y), f_h^{x,y}])||_2}. \tag{4}$$

We concatenate $f_h^{x,y}$ and $l^*(x,y)$ as input to the variation module, $G_{\theta'}$, which we model as a 3-layer convolutional network. $G_{\theta'}$ provides a $d$ dimensional perturbation vector which models the intra-category variation and transforms the categorical code to an instance level code.

**Decoding and Rendering**    Given the localized instance codes for a scene, we render it in the novel view and compare with the ground truth using $L2$ pixel loss. Intuitively, object categories which encode geometric and visual patterns should render the scene more accurately from novel views. Each region of the scene is rendered using an MLP conditioned on the instance codes and the volumetric rendering equation following the convention of NeRF Mildenhall et al. (2021):

$$H_{\hat{\theta}}(l^*_{\text{instance}}(x,y), \mathbf{p}, \mathbf{d}) = (\mathbf{c}, \sigma), \tag{5}$$

Here $\mathbf{p} = (x,y,z)$ is a coordinate in the scene, $\mathbf{d} \in \mathcal{R}^3$ is a view direction, $\mathbf{c}$ is the RGB value at $\mathbf{p}$ in the direction of $\mathbf{d}$ and $\sigma$ is the volume density at that point. Recall that $(x,y,z)$ corresponds to $(x,y)$ in the input frame $I_h$. We can project $(x,y,z)$ into the camera coordinates of the novel view $I'_h$ to get $(x',y')$. This pixel $(x',y')$ in the novel view corresponds to $(x,y)$ in the input frame, meaning they represent the same point in world coordinates. To get an RGB value for $(x,y)$, we use volume rendering along the ray from camera view $I_h$ into the scene, given by

$$\hat{\mathbf{C}}(\mathbf{r}) = \int_{t_n}^{t_f} T(t) \cdot \sigma(t) \cdot \mathbf{c}(t) \cdot dt, \tag{6}$$

where $T(t) = \exp\left(-\int_{t_n}^{t} \sigma(s) \cdot ds\right)$ models absorbance and $t_n$ and $t_f$ are the near and far field. Given a target view with pose $\mathbf{P}$, the ray to the target camera is given by $\mathbf{r}(t) = \mathbf{o} + t \cdot \mathbf{d}$ where $\mathbf{d}$ is a unit direction vector which passes through $(x,y)$. The volume rendering for a particular pixel occurs along this ray. Let $\mathbf{d}'$ be the direction associated with the novel view $(x',y')$ and $\mathbf{r}'(t) = \mathbf{o} + t \cdot \mathbf{d}'$. The pixel intensity at $(x',y')$ is given by $\hat{\mathbf{C}}' = \hat{\mathbf{C}}(\mathbf{r}')$ and our final loss is

$$\mathcal{L}(I_h, I'_h, x, y) = \|\hat{\mathbf{C}}' - I'_h(x',y')\|_2 + s, \tag{7}$$

where $I'_h(x',y')$ is the ground-truth pixel value at $(x',y')$. We penalize the scoring parameter, $s$, from section 3 in the loss to encourage learning a minimal number of codes.

**NRC for Downstream Tasks**    Once the encoder, codebook, and MLP have been trained, we evaluate the learned representation on various downstream tasks. To perform segmentation, we process each image through the trained encoder, $F_\theta$, to obtain the spatial feature map. Each feature vector, $f_{x,y}$ in the spatial map is assigned to the nearest categorical code, $l^*$ in the learned codebook. The categorical codes are then designated to the corresponding pixel to obtain a segmentation mask.

Traditionally in object navigation, frames from the embodied agent are processed by a frozen, pretrained network. The resulting feature vector is then passed to a policy network which chooses an

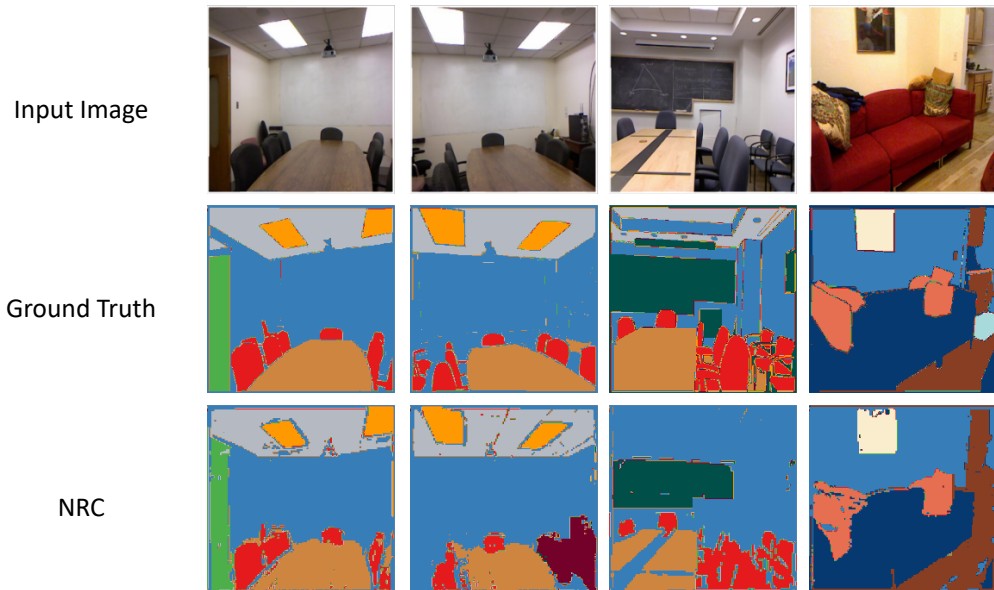

Figure 3: **Unsupervised segmentation of real-world images.** NRC segments scenes that have significant object category overlap with ProcTHOR. We show the first results for object-centric unsupervised segmentation of real-world scenes.

action. To assess the utility of the NRC representation, we replace the pretrained network with the NRC encoder and codebook. We process each frame to obtain instance codes for each region of the image which are then fed to the policy network.

Depth ordering task consists of predicting which of two objects is closer to the camera. To perform depth ordering with NRC we predict a segmentation mask and depth map. To predict the depth map we condition the trained MLP on the instance codes & predict the density, $\sigma$, along a given ray. We estimate the transmittance to predict the depth following the method of Yu et al. (2021a). Depth map and segmentation mask are combined to predict the average distance of each object from the camera.

## 4 EXPERIMENTS

We evaluate our decomposition and representations on several downstream tasks: unsupervised segmentation (real and synthetic), object navigation, and depth ordering. NRC shows improvement over baseline methods on all three tasks. Prior works in object-centric learning have focused on unsupervised segmentation for measuring the quality of their decomposition. We show that NRC representations are also effective for downstream applications that require geometric and semantic understanding of scenes such as object navigation and depth ordering.

### 4.1 DATASETS

**ProcTHOR & RoboTHOR** THOR (Kolve et al., 2017) consists of interactive home environments built in the Unity game engine. We benchmark on the task of object navigation in RoboTHOR (Deitke et al., 2020), a variant of the THOR environment aimed at sim2real benchmarking. Object navigation consists of an agent moving through different scenes to locate specified objects. RoboTHOR consists of 89 indoor scenes split between train, validation, and test. ProcTHOR (Deitke et al., 2022) consists of procedurally generated indoor scenes similar to RoboTHOR. Examples of THOR scenes can be found in Appendix C.

**CLEVR-3D** CLEVR-3D (Johnson et al., 2017) is a synthetic dataset consisting of geometric primitives from multiple views and is used for unsupervised segmentation. Following the convention of Stelzner et al. (2021), we test on the first 500 scenes of the validation set and report foreground-adjusted random index (FG-ARI). Adjusted random index (ARI) Yeung & Ruzzo (2001) measures

the agreement between two clusterings and is a standard metric for unsupervised segmentation. In our case the two clusterings are the predicted and ground truth segmentations. Foreground adjusted random index only measures the ARI for pixels belonging to foreground objects. For comparison to prior works, we consider segmentations at both the class and instance level to be correct for CLEVR-3D, ProcTHOR, and NYU Depth. Further details can be found in Appendix C.

**NYU Depth** The NYU Depth Dataset (Silberman et al., 2012) consists of images from real-world indoor scenes accompanied by depth and segmentation maps. Methods are trained on the ProcThor dataset then evaluated on NYU Depth for segmentation. We chose NYU Depth because it has object categories and scene layouts that are similar to THOR. We report the adjusted random index (ARI).

Table 1: Segmentation results (ARI) for NRC and comparable methods. We find that for more complex datasets, ProcTHOR and NYU Depth, NRC outperforms other methods.

| Method | ProcTHOR (ARI) | NYU Depth (ARI) | CLEVR-3D (FG-ARI) |
|---|---|---|---|
| MarioNette | .127 | .035 | - |
| uORF | .193 | .115 | .962 |
| ObSuRF | .228 | .141 | **.978** |
| NRC | **.295** | **.182** | .977 |

## 4.2 Unsupervised Segmentation

**Experimental Setup** We evaluate NRC, ObSuRF, uORF, and MarioNette for unsupervised segmentation on ProcTHOR, CLEVR-3D, and NYU Depth. We compare with MarioNette because it uses a similar code mechanism for reconstruction. We report FG-ARI on CLEVR-3D for comparison to prior works and ARI on the other datasets. For NYU Depth evaluation we use the representations trained on ProcTHOR and only consider classes that are seen in the training dataset.

**Results** We find that for NYU Depth and ProcTHOR which have more complex layouts and object diversity, NRC significantly outperforms other methods (Table 1). Figure 1 shows examples of the object codes learned by ProcTHOR and Figure 3 shows segmentation examples of real-world images. To our knowledge, this is the first object-centric method which has shown unsupervised segmentation results for complex real-world images. We find that NRC categorizes similar objects across scenes based on both geometry and visual appearance. In the top row of Figure 1, we find that couches of similar shape are assigned to the same code despite differing visual appearance which indicates that NRC codes capture geometric similarity. In the same figure, we show further examples where floors of different texture are categorized by the same code. The improved segmentation performance and geometric categorization of objects indicates that NRC can leverage more than simple color cues which has been significant limitation for object-centric learning.

## 4.3 Object Navigation

**Experimental Setup** We design the object navigation experiments in THOR to understand how well the learned representations transfer from observational data to embodied navigation (Anderson et al., 2018; Batra et al., 2020). Object navigation consists of an embodied agent with the goal of moving through indoor scenes to specified objects. The agent can rotate its camera and move in discrete directions. At each step, agent is fed with the current RGB frame relayed by the camera.

For the representation learning component of the experiment we collect observational video data from a heuristic planner (Appendix B), which walks through procedurally generated ProcTHOR scenes. In total, the dataset consists of 1.5 million video frames from 500 indoor scenes. For further dataset details and example videos see Appendix C.

After training on ProcThor videos, we freeze the visual representations following standard practice (Khandelwal et al., 2022). We train a policy using DD-PPO (Wijmans et al., 2019) for 200M steps on the training set of RoboTHOR then evaluate on the test set. We report success rate (SR) and success weighted by path length (SPL). Success is defined as the agent signaling the stop action within 1 meter of the goal object with it in the view. SPL is defined as $\frac{1}{N} \sum_{i=1}^{N} S_i \frac{\ell_i}{\max(p_i, \ell_i)}$, where $l_i$ is the shortest possible path, $p_i$ is the taken path, & $S_i$ is the binary indicator of success for episode $i$.

Table 2: Results for object navigation on RoboTHOR object navigation. Visual representations are trained on observations from 500 scenes of ProcThor. A policy is learned on top of the frozen visual representations by training on the object navigation task in RoboTHOR training scenes. The results are obtained by evaluating on RoboTHOR test scenes.

| Method | Success Rate (%) | SPL |
|---|---|---|
| uORF (Yu et al., 2021b) | 31.3 | .146 |
| ImageNet Pretraining | 33.4 | .150 |
| ObSuRF (Stelzner et al., 2021) | 38.9 | .167 |
| Video MoCo (Feichtenhofer et al., 2021) | 43.9 | .184 |
| EmbCLIP (Khandelwal et al., 2022) | 47.0 | .200 |
| NRC (Ours) | **50.1** | **.239** |

We compare with the following baselines: ObSuRF, uORF, Video MoCo (Feichtenhofer et al., 2021), and EmbedCLIP (Khandelwal et al., 2022). ObSuRF and uORF are 3D, object-centric methods, and Video MoCo is a contrastive video representation learning method. We include Video MoCo for comparison as it was designed for large-scale, discriminative tasks, while ObSuRF and uORF were primarily intended for segmentation. See Appendix B for further implementation details of baselines.

**Results**   Table 2 shows the performance of NRC and baselines on RoboTHOR object navigation. NRC outperforms the best baseline by 3% in success rate (SR) and by a 20% relative improvement in SPL. These performance gains indicate that the geometrically-aware NRC representation provide an advantage over traditional representations such as EmbCLIP and Video MoCo. Another key observation is that NRC has at least 18.8% improvement in SR and relative SPL over the recent object-centric learning methods, uORF and ObSuRF. In particular, NRC performs better than ObSuRF and uORF when navigating near furniture and other immovable objects (see supplemental for object navigation videos). We hypothesize this is due to NRC's more precise localization of objects.

## 4.4   DEPTH ORDERING

**Experimental Setup**   We evaluate NRC on the task of ordering objects based on their depth from the camera. Understanding the relative depth of objects requires both geometric and semantic understanding of a scene. For this task we evaluate on the ProcTHOR test dataset which provides dense depth and segmentation maps. Following the convention of Ehsani et al. (2018), we determine ground truth depth of each object by computing the mean depth of all pixels associated with its ground truth segmentation mask.

For evaluation, we select pairs of objects in a scene and the goal is to predict which object is closer. We take the segmentation that has the largest IoU with the ground truth mask as the predicted object mask. To determine the predicted object depth, we compute the mean predicted depth of each pixel associated with the predicted object mask. All representations are trained on the ProcTHOR dataset and evaluated on the ProcTHOR test set. In total we evaluate 2,000 object pairs.

Table 3: We compare depth ordering results on RoboTHOR with other geometrically-aware representations. Given pairs of objects in the scene, the model must infer which object is closer. We report the accuracy as the number of correct orderings over the total number of object pairs.

| Method | Depth Order Acc. (%) |
|---|---|
| uORF (Yu et al., 2021b) | 13.5 |
| ObSuRF (Stelzner et al., 2021) | 18.3 |
| NRC (Ours) | **23.8** |

**Results**   Depth ordering requires accurate segmentation and depth estimation. Due to NRC's stronger segmentation performance and better depth estimation, we see 5.5% and 10.3% depth ordering accuracy compared to ObSuRF and uORF respectively. The fine-grained localization of categorical latent codes in NRC allows for better depth ordering over existing object-centric methods. In particular, the other object-centric methods tend to assign object instances to the background which leads to large errors in estimating the depth of objects (Figure 5).

## 4.5 ABLATION STUDY

Table 4: Ablation study for modeling the intra-code variation and learning the number of codes evaluated on unsupervised segmentation in ProcTHOR. Default fixed number of codes is set to 25.

| Method + (Ablation) | ProcTHOR (ARI) |
|---|---|
| NRC | .182 |
| NRC + Learned # of Codes | .197 |
| NRC + Variation Module | .284 |
| NRC + Variation Module + Learned # of Codes | **.295** |

We present an ablation study on the ProcTHOR dataset to determine the effect of the variation module and learnable codebook size on unsupervised segmentation performance. Quantitative results can be found in Table 4. We observe that performance improves by $\sim 9\%$ when intra-class variation is explicitly modeled. Intuitively, allowing for small variation between instances of the same category should lead to better representations and allow for greater expressiveness.

We also find that learning the number of codes moderately improves performance. However, if number of codes is found via hyper-parameter tuning, performance is matched (Table 8). Nonetheless, differentiably learning the codebook size avoids computationally expensive hyper-parameter tuning.

## 5 LIMITATIONS

Novel view reconstruction requires camera pose, which is not available for most images and videos. Some datasets such as Ego4D (Grauman et al., 2022) provide data from inertial measurement units that can be used to approximate camera pose, although this approach is prone to drift.

An incorrect assumption that NRC and most object-centric prior works make is that scenes are static. However, it rarely is the case that scenes are free of movement due to the physical dynamics of our world. Recently, Kipf et al. (2021); Pumarola et al. (2021) made strides in learning representations from dynamic scenes.

Although NRC is relatively efficient compared to the other NeRF based methods, the NeRF sampling procedure is compute and memory intensive. Sajjadi et al. (2022a) and Smith et al. (2022) leverage object-centric light fields to reduce memory and compute costs. The efficiency improvement from modeling scenes as light fields is orthogonal to NRC and can be combined.

A final challenge inherent to novel view reconstruction is finding appropriate corresponding frames of videos. For example, if two subsequent frames differ by a 60° rotation of the camera, then most of the scene in the subsequent frame will be completely new. Therefore, constructing the content in the novel view is ill-posed. Pairing frames with overlapping frustums is a potential solution, although the content of the scene may not be contained in the intersecting volume of the frustums.

## 6 CONCLUSION

Compositional, object-centric understanding of the world is a fundamental characteristic of human vision and such representations have the potential to enable high-level reasoning and efficient transfer to downstream tasks. Towards this goal, we presented Neural Radiance Field Codebooks (NRC), a new approach for learning geometry-aware, object-centric representations through novel view reconstruction. By jointly learning a shared dictionary of object codes through a differentiable renderer and explicitly localizing object codes within the scene, NRC finds reoccurring geometric and visual similarities to form objects categories. Through experiments, we show that NRC representations improve performance on object navigation and depth ordering compared to strong baselines by 3.1% success rate and 5.5% accuracy respectively. Additionally, we find our method is capable of scaling to complex scenes with more objects and greater diversity. NRC shows relative ARI improvement over baselines for unsupervised segmentation by 29.4% on ProcTHOR and 29.0% on NYU Depth. Qualitatively, NRC representations trained on synthetic data from ProcTHOR show reasonable transfer to real-world scenes from NYU Depth.

ACKNOWLEDGEMENTS

We thank Kuo-Hao Zeng for technical advice regarding the THOR environment, Reza Salehi and Vishwas Sathish for helpful discussion, and Koven Yu and Mehdi Sajjadi for prompt correspondence regarding details of their methods. This work is in part supported by NSF IIS 1652052, IIS 17303166, DARPA N66001-19-2-4031, DARPA W911NF-15-1-0543 and gifts from Allen Institute for Artificial Intelligence and Google.

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

## A    FURTHER EXPERIMENTS AND QUALITATIVE EXAMPLES

### A.1    MORE COMPLEX REAL-WORLD SCENES (NYU-DEPTH)

We provide more segmentation examples of NRC on cluttered, real-world scenes (figure 4). We find that NRC reasonably segments categories that overlap with those of THOR. Additionally we evaluate NRC and comparable methods on more difficult scenes of NYU Depth to see how complexity of the scene affects performance. Specifically we filter for scenes with 5 or more objects in them for evaluation on unsupervised segmentation. In general performance degrades (Table 5), though NRC performance decreases less compared to uORF and ObSuRF.

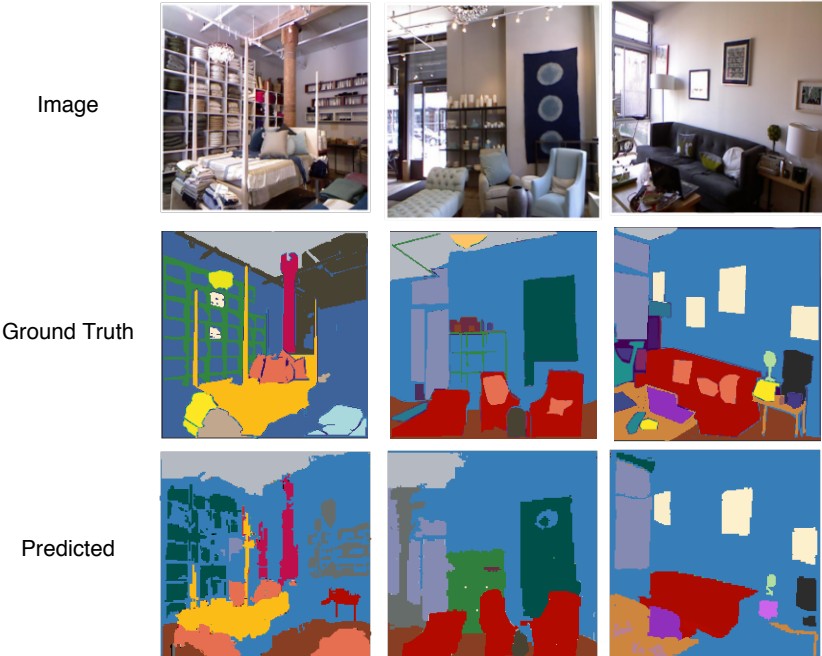

Figure 4: **Segmentation examples of cluttered scenes.** Further segmentation examples of cluttered, real-world scenes from NYU Depth. NRC scales better to cluttered scenes compared to other methods as the rendering and memory cost is constant in the number of objects.

### A.2    MATTERPORT-3D AND GIBSON EXPERIMENTS

We train NRC on scenes from MatterPort3D. Qualitative examples can be found in figure 6 and quantitative results can be found in Table 6. We find NRC performs slightly worse on MatterPort3D compared to on THOR likely due to the greater visual complexity. This is reinforced by the evidence that the learned codebook size is 67 whereas for THOR it is on average $\sim 53$. A greater number of learned codes indicates that NRC requires more expressivity to accurately reconstruct Matterport3D scenes. For a given scene we find corresponding frames by filtering for images within 20 degrees viewing angle of each other.

We evaluate NRC and ObSuRF on the Habitat Point Navigation Challenge. We train NRC and ObSuRF on data collected from random walks through the Habitat environment. We use a ResNet-50 initialized with a MoCo backbone pretrained on ImageNet. We report success rate, shortest-path length, and distance to goal. Qualitative examples can be found in the supplemental. We find that NRC outperforms ObSuRF and ImageNet pretraining. Qualitatively we find that NRC is better at navigating around objects compared to ObSuRF.

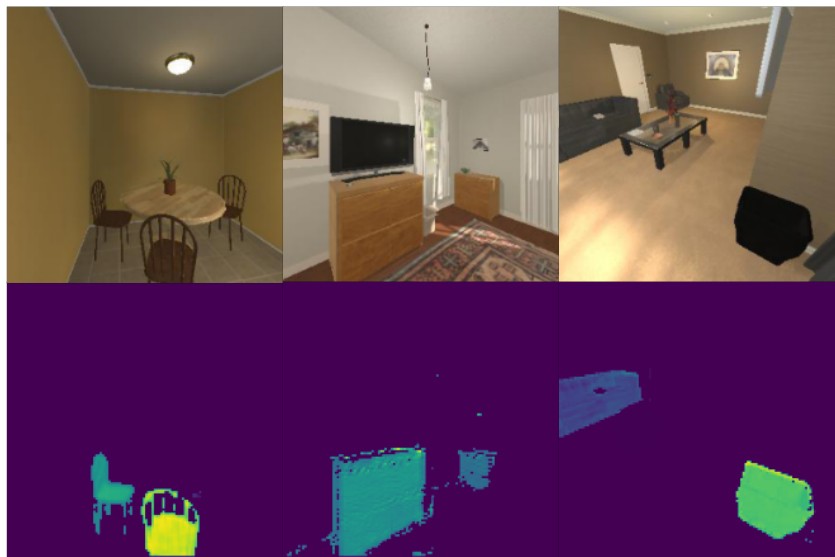

Figure 5: **Predicted object depth maps for depth ordering.** The predicted object masks are overlayed with the predicted depth map. We average the per pixel depth for each predicted object mask to infer which object is closer. NRC's object localization leads to sharp object masks and less error when predicting average object depth.

## B  IMPLEMENTATION DETAILS

### B.1  NRC

For the segmentation experiments we use the modified ResNet34 used by Yu et al. (2021b). For the decoder we use a 3-layer with hidden dimension of 512. We train with a learning rate of $1e-4$ with the ADAM optimizer. We set the near field to .4 and far field to 5.5. We train from scratch, and from the video MoCo model trained on ProcTHOR. We found starting from the pretrained initialization reduced the number of training epochs required for convergence. For the model from scratch we train for 300 epochs. For the model from MoCo initialization we train for 100 epochs. While training we use a sliding window of 5 frames to determine corresponding images in ProcTHOR. By corresponding images we mean the image given to the model and the ground truth novel view image. Given a sliding window of n frames we randomly select 2 images from this interval. We ablate over 2, 5, and 10 frames. We found 5 images led to the best segmentation results and therefore used that window size for object-navigation.

For object-navigation we use a ResNet50 for fair comparison to CLIP and Video MoCo. We train wit the same hyper-parameters as discussed above for segmentation with the only change being the architecture. We train for 200M steps using PPO with default hyper-parameters to match Khandelwal et al. (2022) in RoboTHOR.

Table 5: **Performance on cluttered scenes of NYU Depth.** We filter for scenes with 5 or more objects and report the ARI of NRC and comparable methods. NRC outperforms ObSuRF and uORF particularly on cluttered scenes, as it can support an unbounded number of objects.

| Method | NYU Depth (Hard) |
|--------|------------------|
| uORF   | .046 |
| ObSuRF | .092 |
| NRC    | .139 |

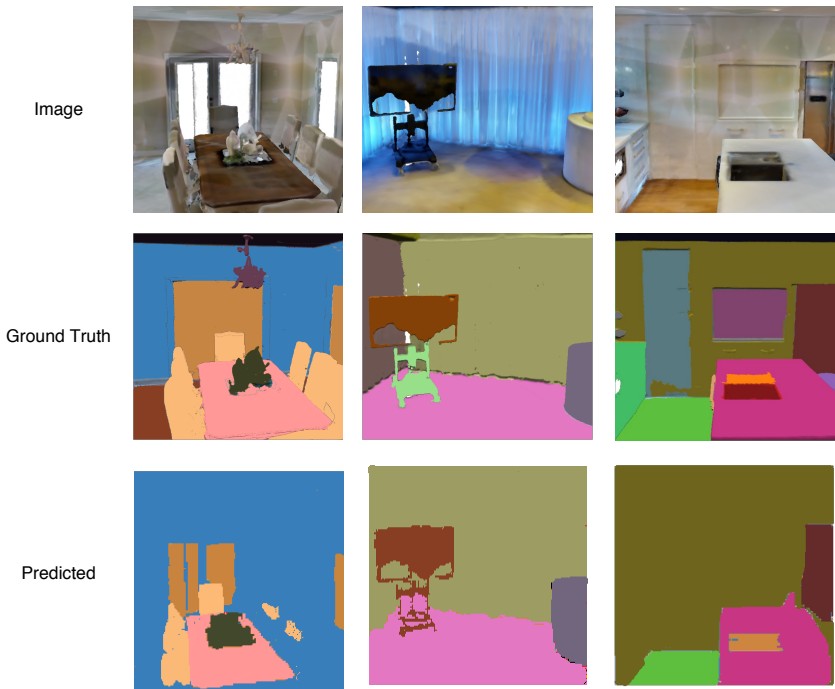

Figure 6: **Segmentation examples for MatterPort3D.** Segmentation examples on MatterPort3D. We train NRC on ∼ 190,000 images of MatterPort3d and learn 67 categorical codes. This increase in the learned number of codes is likely due to the increased visual complexity of MatterPort3D compared to THOR. We find similar segmentation results to those on THOR and NYU Depth.

## B.2 UORF

We use the implementation available at `https://github.com/KovenYu/uORF` We train on the training set of ProcTHOR which contains ∼ 1.5 million images. We train three different variations of the uORF no GAN model: from scratch, fine-tune with encoder and decoder pre-trained on CLEVR-567, and fine-tune with all three components pre-trained on CLEVR-567. For variations that train the slot attention component from scratch, we try setting the number of slots to 8 (default) and 16. We train for 10 epochs when loading from a pre-trained model, and for 15 epochs when training from scratch; roughly half of the epochs are coarse epochs, and we set percept_in to roughly one third of the epochs. We train for less epochs because our dataset is much larger. Our images within scene batch size is 4, and we turn off shuffling to ensure that batches contain images that are close together within the scene. We pass in the fixed locality parameter, but also set the decoder's locality parameter to false. To compensate between differences between our dataset and CLEVR-567, we tune nss scale, object scale, near plane, and far plane. We train models on a single NVIDIA A40.

## B.3 OBSURF

We use the official implementation provided by Stelzner et al. (2021) at `https://github.com/stelzner/obsurf`. We trained 3 variations of the ObSuRF model: onfrom scratch, from CLEVR-

Table 6: **Unsupervised segmentation performance of NRC and ObSuRF on MatterPort3D.** We find NRC performs slightly worse on MatterPort3D compared to on THOR likely due to the greater visual complexity. This is reinforced by the observation that NRC learns 67 codes for MatterPort3D whereas 53 are learned for THOR.

| Method | MatterPort3D |
| --- | --- |
| ObSuRF | .178 |
| NRC | .237 |

Table 7: **Habitat Point Navigation Challenge.** The performance of various visual encoders on the point navigation challenge. We find that NRC outperforms other 3D-object centric methods due to its ability to scale to more objects and visually complex environments.

| Method | SPL | SR | Goal Dist |
|---|---|---|---|
| ImageNet Pretrain | .82 | .94 | .73 |
| ObSuRF | .77 | .84 | .81 |
| NRC | .85 | .96 | .68 |

3D pretrained weights, and from MultiShapeNet pretrained weights. For the pretrained models we lowered the learning rate by a factor of 10 to $1e^-5$. For training from scratch we used the default learning rate of $1e - 4$. For the number of slots we tried the default number, 5, and 10. We report the results of the best model which was from scratch with 10 slots. For object navigation experiments in RoboTHOR, we used the model with the highest ARI which was training from scratch with 10 slots. We learn a policy network comprised of a 1-layer GRU with 512 hidden units which maps to a 6 dimensional logit and scalar which are used for the actor-critic. We train with DD-PPO for 200M steps. For the encoder, we use a ResNet-50 architecture.

To construct the feature vector fed to the GRU we use the output of the ResNet-50 which is $2048 \times 7 \times 7$. We pass this through a 2-convolutional layers to sample down to $32 \times 7 \times 7$ then concatenate this with the trainable goal state embedding matrix which is $32x7x7$. This visual feature vector of shape $64 \times 7 \times 7$ is then passed through another 2-layer convolutional network to be of size $32 \times 7 \times 7$.

### B.4 OSRT

We do not compare with OSRT Sajjadi et al. (2022a) on data sets that were not evaluated in the original paper. We contacted the authors, but a public implementation has not been released yet at the time of this work.

## C  DATASETS

### C.1  PROCTHOR

The ProcTHOR dataset Deitke et al. (2022) consists of 10,000 procedurally generated indoor rooms in the THOR environment. For our dataset we collect 10 video sequences of 300 actions from 500 randomly chosen training scenes. Each actions consisting of either rotation or stepping in a specified direction. Actions are determined by a heuristic planner which moves throughout the scene. In total we collect 1.5 millions image frames. See Figure 7 for a sample of the dataset.

For evaluation of ARI we consider the 20 largest objects, floor, and ceiling. uORF and ObSuRF decompose scenes into 5-10 objects and therefore do not handle scenes with many small objects. We evaluate ARI only on the 20 largest objects in THOR which are the following: {ArmChair, Bathtub, BathtubBasin, Bed, Cabinet, Drawer, Dresser, Chair, CounterTop, Curtains, Desk, Desktop, CoffeeTable, DiningTable, SideTable, Floor, Fridge, Television, TVStand, Toilet.}

### C.2  CLEVR-3D

CLEVR-3D (Johnson et al., 2017) is a synthetic dataset which consists of geometric primitives placed on a monochrome background. Following the convention of (Stelzner et al., 2021) we evaluate on this benchmark for unsupervised object segmentation. Following convention we evaluate on the first 500 scenes of the validation set and report foreground - adjusted random index. For class level labels we consider objects with the shape to be of the same class.

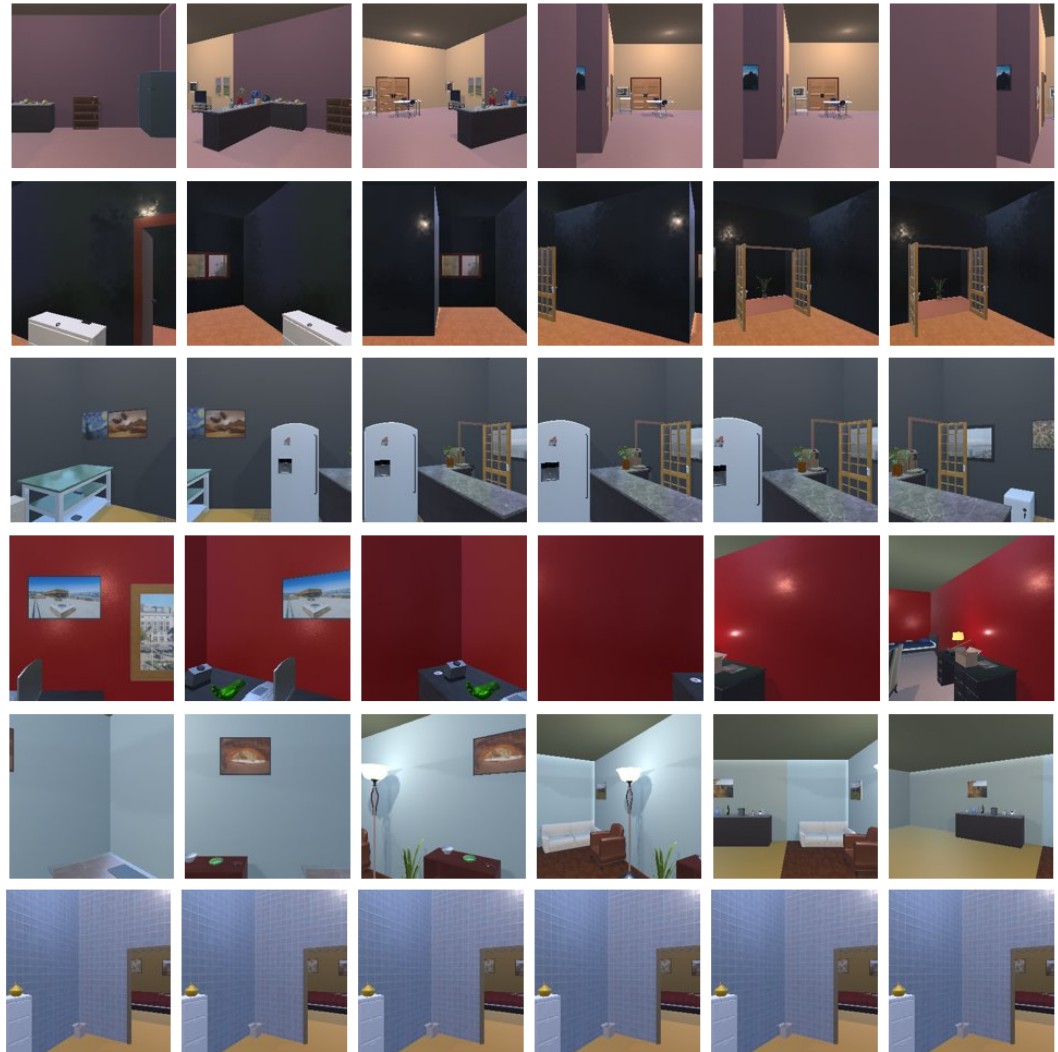

Figure 7: Snippets of Videos from ProcTHOR dataset used for training NRC.

Table 8: Performance of NRC with a fixed codebook of various sizes on ProcTHOR.

| Codebook Size | ProcTHOR (ARI) |
|---|---|
| 10 | .247 |
| 20 | .256 |
| 30 | .266 |
| 40 | .279 |
| 50 | .286 |
| 60 | .293 |
| 70 | .288 |
| 80 | .295 |
| 90 | .279 |
| 100 | .273 |

Input View      Novel View      Predicted

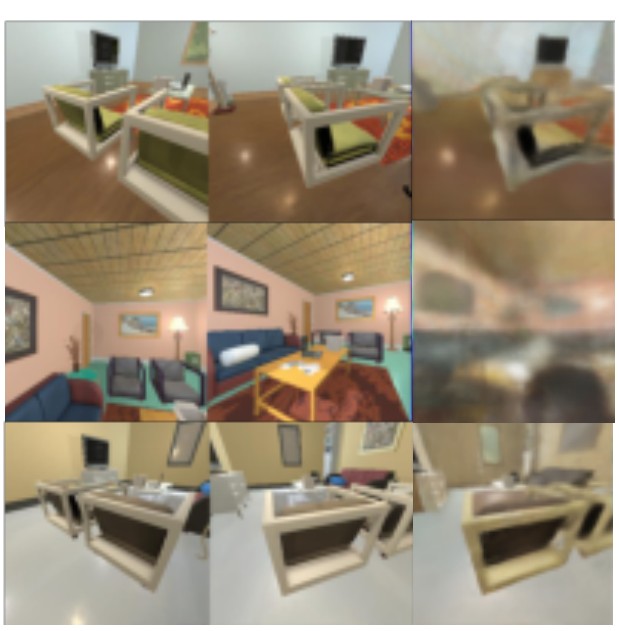

Figure 8: **Reconstruction examples from NRC.** For scenes and objects that are closer to the training distribution, reconstruction quality is significantly better.

### C.3 INSTANCE LEVEL ACCURACY

For comparison to the instance level classification performed by ObSuRF and uORF, we filtered for images in ProcTHOR which have one object of each class. In this setting, class and instance level classification are equivalent.

Table 9: The size of the learned codebook averaged over 5 runs on each dataset.

|  | Codebook Size |
| --- | --- |
| CLEVR-3D | $15.6 \pm 1.2$ |
| ProcTHOR | $53.8 \pm 2.4$ |

