# OpenReview forum: "Neural Radiance Field Codebooks"
_ICLR.cc/2023/Conference — ICLR 2023 poster_

### Official Review · Reviewer_GeqQ · 2022-10-24

**Confidence:** 4
**Correctness:** 3
**Technical Novelty And Significance:** 2
**Empirical Novelty And Significance:** Not applicable
**Recommendation:** 5

**Clarity, Quality, Novelty And Reproducibility:**

Overall, the technical originality is moderate, it extends the idea in MarioNette, with additional new modules. And I can follow the paper successfully, but the exposition could be improved for clarity.

1. Figure 2 is not self-contained and is not consistent with the textual description of the technical part. For example, what is $F$ in the figure, where are the module $H_{\theta}, G_{\theta}$ in the figure.

2. In Figure 1, how does the label of the catogery objects come, say, Couch, Floor, and Fridge? If I follow the paper correctly, the learned dictionary does not provide catogerical labels.

3. In Table 2, what is the ImageNet Pretraining at the second row. Is it a segmentation module pretrained on ImageNet? If yes, it may be unfair to compare on synthetic data.

4. Although the submission includes source code of some experiments, it would be appreciated much that the textual implementation details can be included first in the supplementary, with source code as the strong complementary.


A few spots I found that need correction:
1. In Section 2, "NERF" -> "NeRF"

2. Section 4.1, "between train, valiation, and test" ...




**Strength And Weaknesses:**

Strength:

1. I value the concept of learning of a codebook over 3D scenes, which could benefit many downstream tasks.

2. this paper extends the technical route presented in the MarioNette for unsupervised dictionary learning on 3D scenes, with some novel technical contributions for working on 3D scenes, i.e., the modules for learning instance variation and the dynamic changing number of objects.


Weakness:

My concerns over this paper are mainly on the evaluation/demonstration of the efficacy of the proposed learning setup.

1. the paper ships itself that it learns reoccurring 3D objects across scenes, and explicitly localize the learned objects, but the demonstration of this is only done in a indirect way. Ideally, a successful model should be able to segment the 3D scene into meaningful 3D units, thus the evaluation of the learning could have focused on the 3D, i.e., measuring the cluster of predicted instances/objects against the GT cluster, a similar way as done for the unsupervised image segmenation as show in the paper, but with an emphasis in 3D. Moreover, more visual results of how 3D scenes are decomposed into meaningful units should be presented for qualitative examing the performance.

2. the evaluation are mostly done on synthetic data. While this is not a fatal flaw, there is indeed no enough evidence to support that the proposed method can be applicable on more real scenarios. ScanNet and Matterport3D for example contain 3D scenes collected from the real world, with estimated camera poses available. I am cuorious about the performance on these kinds of real-world data, where the noisy camera estimates, difficulty in finding corresponding frames, and the imaging quality differ significantly from synthetic data.

3. for the object navigation task, again, I think the evaluation can be conducted in a bit more realistic setting. What about evaluating on Gibson and Matterport3D scenes, where we can have more realistic scenes with camera poses available.

**Summary Of The Paper:**

The paper presents a method to learn a shared set of object representations from paired views captured across 3D scenes, in a self-unsupervised manner. The core difference to existing 3D object-centric learning work is that it learns reoccurring objects across scenes and explicitly localize the learned objects, which are demonstrated in a set of downstream tasks, such as unsupervised image segmenation, object navigation, and depth ordering, that require the understanding of the scene layout and object goemetry and appearance.

**Summary Of The Review:**

While I recognize the value of learning a codebook of 3D units over 3D scenes, my concerns expressed above make me leaning towards negative on this paper.

---

> ### Author Response · Authors · 2022-11-12
> **Author Response**
>
> Thank you for your thorough review. We have modified the paper to reflect your feedback and are interested in any further feedback you may have. We have summarized the changes made to the new version of the paper in top-level comment titled “Summarization of Changes”. Below you will find our responses to your question and comments.
>
> **The paper ships itself that it learns reoccurring 3D objects across scenes, and explicitly localize the learned objects, but the demonstration of this is only done in an indirect way. Ideally, a successful model should be able to segment the 3D scene into meaningful 3D units, thus the evaluation of the learning could have focused on the 3D, i.e., measuring the cluster of predicted instances/objects against the GT cluster, a similar way as done for the unsupervised image segmenation as show in the paper, but with an emphasis in 3D. Moreover, more visual results of how 3D scenes are decomposed into meaningful units should be presented for qualitative examing the performance.**
>
> We evaluated our method on unsupervised segmentation as that is the standard task for object-centric works. Most real-world scenes do not have voxel ground truth so for quantitative evaluation we chose 2D segmentation. Implicitly the 3D shape of the object can be visualized by segmenting it from novel views. Based on your feedback we have created videos of 3D object segmentation. Please see the 3D segmentation folder in the supplementary for these examples.
>
> **The evaluation are mostly done on synthetic data. While this is not a fatal flaw, there is indeed no enough evidence to support that the proposed method can be applicable on more real scenarios. ScanNet and Matterport3D for example contain 3D scenes collected from the real world, with estimated camera poses available.**
>
> Thanks for the suggestion. We have trained NRC on Matterport3D and have added the results to the paper. Please see appendix A.2 for quantitative and qualitative unsupervised segmentation results on Matterport3D (figure 6 and table 6). The object navigation portion of the experiment takes multiple days, so we will update the reviewers when those are added.
>
> We would also like to point out that we are the first object-centric method to show unsupervised segmentation results on complex real-world world scenes (NYU-Depth). Previous methods in object-centric learning such as uORF and ObSuRF have been limited to synthetic scenes. While NRC doesn’t address all the challenges of object-centric learning from real-world data, our experiments demonstrate that NRC scales better to complex real-world scenes than existing methods.
>
> **For the object navigation task, again, I think the evaluation can be conducted in a bit more realistic setting. What about evaluating on Gibson and Matterport3D scenes, where we can have more realistic scenes with camera poses available.**
>
> We appreciate the suggestion. We have trained NRC on Matterport3D and are running the object navigation experiments. **Update -** We have run the Habitat experiments with Gibson. Please see our *Author Update* comment below.
>
> **Clarity:**
>
> **Figure 2 is not self-contained and is not consistent with the textual description of the technical part. For example, what is  in the figure, where are the module in the figure.**
>
> Thanks for the feedback. We have updated the figure to be consistent with the notation in the methods section.
>
> **In Figure 1, how does the label of the category objects come, say, Couch, Floor, and Fridge? If I follow the paper correctly, the learned dictionary does not provide categorical labels.**
>
> You are correct, the codebook does not receive semantic labels. We added semantic label to the figure so that readers can more easily parse what categories of objects are being assigned to the learned codes.
>
> **In Table 2, what is the ImageNet Pretraining at the second row. Is it a segmentation module pretrained on ImageNet? If yes, it may be unfair to compare on synthetic data.**
>
> ImageNet pretraining is a standard baseline in object-navigation so we included it in the table for reference. The comparable methods which are trained on the same data are uORF, ObSuRF, and Video MoCo.
>
> **Although the submission includes source code of some experiments, it would be much appreciated that the textual implementation details can be included first in the supplementary, with source code as the strong complementary.**
>
> Thanks for the suggestion. We have added implementation details for NRC to appendix B.1.
>
> **Corrections:**
> - **In Section 2, "NERF" -> "NeRF"**
>
> - **Section 4.1, "between train, valiation, and test" ...**
>
> Thanks. We have made these corrections accordingly.

---

> > ### Author Response · Authors · 2022-11-18
> > **Author Update**
> >
> > Given your recommendation to evaluate NRC on Gibson and Matterport-3D scenes, we have trained NRC on the Habitat Point Navigation Challenge. We chose point navigation as the moderate training time allowed us to evaluate other methods for comparison (ObSuRF and ImageNet Pretrain). Our results can be found in table 7 of the revised appendix and qualitative examples can be found in the *Object Navigation/gibson* folder. We've added details and analysis in the "Matterport-3D and Gibson Experiments" section of the appendix. We found that NRC outperformed ObSuRF and ImageNet pretraining in both shortest path length and success rate. Qualitatively we found NRC was better at navigating in tight spaces and near other objects.
> >
> > | Method            | SPL | SR  | Goal Dist |
> > |-------------------|-----|-----|-----------|
> > | ObSuRF            | .77 | .84 | .81       |
> > | ImageNet Pretrain | .82 | .94 | .73       |
> > | NRC               | .85 | .96 | .68       |

---

### Official Review · Reviewer_BHLx · 2022-10-24

**Confidence:** 3
**Correctness:** 3
**Technical Novelty And Significance:** 2
**Empirical Novelty And Significance:** 2
**Recommendation:** 5

**Clarity, Quality, Novelty And Reproducibility:**

The method is not clearly demonstrated as several critical parts are missing. The novelty is limited as the dictionary learning part is mainly borrowed from Marionette. The experimental results are not insufficient to verify the efficacy of the learned scene representation.

**Strength And Weaknesses:**

 Pros:
+ The major differences between the proposed method and others are clearly presented and easy to follow.
+ The paper tries to address a critical problem of learning object-centric neural scene representation

Cons:
- Several key elements of the proposed method are missing. For instance, how the codebook is learned should be clarified even if it is similar to Marionette. Otherwise, it is hard to understand how the codebook in Fig. 2 is constructed during training. Besides, how the learned object-centric representation is applied to the three tasks should be detailed with illustrations.
- The experimental results provide limited information regarding the performance of the proposed method. Without qualitative results of object navigation (video for instance) and depth ordering tasks, readers cannot intuitively understand how the proposed method performs. The analysis for results on view synthesis, unsupervised segmentation, object navigation, and depth ordering is also insufficient.

Minors:
1. The floor in Fig. 1 is placed in the middle row instead of the "bottom row"
2. "Additionally, we learn can model ..." (page 3)
3. The reused annotations lead to confusions: i \in n indicates the index of scenes, while i \in k indicates the index of codes; (x,y) indicates the pixel position while (x,y,z) indicates the world coordinate.  (page 3)
4. The symbols such as O_h, f_h^{x,y}， G_\theta, H_\theta are encouraged to be added to Fig. 2. It is unclear where the concatenation of the relative position and the absolute position occur in Fig. 2.



**Summary Of The Paper:**

The paper target the learning of an object-centric scene representation through NeRF-wise volume rendering. The key component is inspired by Marionette (Smirnov et al., 2021) to utilize a dictionary of object codes. The modifications include a gated function that allows dynamic addition of new categorical codes, category-wise code to instance-wise code through a variation module. Experiments indicate that the learned representation can be applied to tasks such as unsupervised segmentation, object navigation, and depth ordering.

**Summary Of The Review:**

 Due to the issues mentioned in the Weakness section, I am leaning toward rejection currently. Though the experimental results in Tab. 4 indicate that the modification of the proposed method does bring benefits, major revision is required to make the method and the corresponding applications clear and intuitive. I would like to see how the authors respond and the opinions from other reviewers.

---

> ### Author Response · Authors · 2022-11-12
> **Author Response**
>
> Thank you for your thorough review. We have modified the paper to reflect your feedback and are interested in any further feedback you may have. We have summarized the changes made to the new version of the paper in top-level comment titled “Summarization of Changes”. Below you will find our responses to your question and comments.
>
> **Several key elements of the proposed method are missing. For instance, how the codebook is learned should be clarified even if it is similar to Marionette. Otherwise, it is hard to understand how the codebook in Fig. 2 is constructed during training. Besides, how are learned object-centric representations applied to the three tasks.**
>
> Thank you for the feedback. Overall we have revised the methods section to improve clarity based on your feedback. We have changed the section *Assigning Categorical Codes* to *Assigning + Learning Categorical Codes* and added the missing details about how the codebook is learned. To address your specific question, the codebook is learned by using soft nearest neighbors with the straight-through-estimator (STE) during the backward pass. This allows gradients to propagate to the codebook even though the hard nearest-neighbors assignment on the forward pass is non-differentiable.
>
> To clarify how NRC is applied to the three tasks we have added a subsection in the methods, *Using NRC for Downstream Tasks*. The subsection explains how the codebook is used for segmentation, object-navigation, and depth ordering. We briefly summarize below:
>
> - Segmentation - We process each image through the trained encoder to obtain the spatial feature map. For each feature vector in the spatial map we assign the nearest categorical code. The categorical codes are designated to the corresponding pixel to obtain a segmentation mask.
> - Object navigation - The standard pipeline for object navigation is a pretrained convolutional network followed by a policy network. For NRC we replace the convolutional network with the NRC encoder and codebook and pass the output of the codebook assignment to the policy network.
> - Depth Ordering - We predict the segmentation mask and depth map for a given image. We use the MLP conditioned on the latent codes to estimate the transmittance function along a ray and threshold it to obtain the depth. The depth map and segmentation mask are combined to predict the average distance of the object from the camera.
>
> **The experimental results provide limited information regarding the performance of the proposed method. Without qualitative results of object navigation (video for instance) and depth ordering tasks, readers cannot intuitively understand how the proposed method performs. The analysis for results on view synthesis, unsupervised segmentation, object navigation, and depth ordering is also insufficient.**
>
> Thank you for the feedback. We have added videos of the agent performing object-navigation to the supplemental (object navigation folder) and qualitative examples for depth ordering and view synthesis to appendix A (figure 5 and 8).  We added additional analysis for each downstream task to the results subsection in experiments. Additionally we have added further qualitative examples and analysis for unsupervised segmentation on Matterport3D and NYU Depth (figure 4, figure 6, table 5). Please see the updated revision and supplemental for the new content.
>
> **Minor:**
> - **The floor in Fig. 1 is placed in the middle row instead of the "bottom row"
> "Additionally, we learn can model ..." (page 3)**
> - **The reused annotations lead to confusions: $i \in n$ indicates the index of scenes, while $i \in k$ indicates the index of codes.**
> - **The symbols such as $O_h$, $f_h^{x,y}$， $G_\theta$, $H_\theta$ are encouraged to be added to Fig. 2. It is unclear where the concatenation of the relative position and the absolute position occur in Fig. 2.**
>
> Thanks for the detailed feedback, we’ve revised the paper accordingly to reflect your recommendations.

---

### Official Review · Reviewer_77Ao · 2022-10-24

**Confidence:** 4
**Correctness:** 3
**Technical Novelty And Significance:** 3
**Empirical Novelty And Significance:** 3
**Recommendation:** 8

**Clarity, Quality, Novelty And Reproducibility:**

### Clarity

Most of the paper was easy to follow, with the exception I mention above.

---

### Quality

I raised 3 questions above regarding some gaps in the method.

---

### Novelty

There are some partially similar works, but this is a field with a **very** high publication rate, so this is not a reason to reject.
Specifically, here are 3 similar works, that can be mentioned in the text:
1. [In-Place Scene Labelling and Understanding with Implicit Scene Representation](https://shuaifengzhi.com/Semantic-NeRF/) [ICCV 2021]
2. [NeRF-SOS: Any-View Self-supervised Object Segmentation on Complex Scenes](https://zhiwenfan.github.io/NeRF-SOS/) [TMLR 2022]
3. [NeSF: Neural Semantic Fields for Generalizable Semantic Segmentation of 3D Scenes](https://nesf3d.github.io/) [arXiv]

---

### Reproducibility

The method itself is quite straightforward and should be possible to implement over an existing version of NeRF.

**Strength And Weaknesses:**

### Strength
The manuscript is generally well written, and the method seems rather straightforward.
The experimental section presents impressive transfer from synthetic to real images.

### Weaknesses
There are issues I hope the authors can clarify:

1. It seems that the input, as in PixelNeRF, images are needed in inference time, otherwise - where is the feature map obtained from to query the dictionary?
If this is so - this is a major caveat compared to other NeRF methods that can render an image using only the MLP.

2. Another possible discrepancy is the fact that the atom is generated from 2D coordinates of a view different from the novel one - but unlike "PixelNeRF", if an object is occluded in the "input view" we will get **a wrong atom** and hence the wrong class for the segmentation map - is this resolved by a low density for the rendering process as well?

3. How exactly is a new atom added to the dictionary? Specifically, when the parameter 's' "signifies when to add a new code" - what value is the atom given?

There are some places in the text that can be made clearer:
- I would recommend using a different notation for the two input images, rather than h and h'.
- Figure 2 - It is not clear which part of the pipeline that are included in the MLP training, are not included in inference. For example:
   - Where is the feature map F generated from?
   - What feature values (or atoms) will we get for queries of regions out of the "input view"?
- In addition, it was hard to follow how eq2 is incorporated into the flow - are discarded atoms kept as zero?


**Summary Of The Paper:**

This paper aims to marry the power of semantic codebooks with NeRFs, by conditioning the MLP on a fixed codebook (fixed after training, of course).

We are given two views - "input view" A and "novel view" B, as well as a shared codebook
The first stage is to run A through an encoder network, obtaining the (2D!) feature map F.
Next, each 3D query point needed to render B is projected into A's 2D coordinates, essentially retrieving the relevant "pixel feature" f from F. We find the nearest neighbor atom m in the dictionary of the aforementioned "pixel feature" f, and add a small perturbation to account for intra class variability. This perturbation is generated by feeding both m and f into a small MLP.
The final value (atom+pertubation) is concatenated to the standard NeRF inputs, that is the view direction and the aforementioned 3D query point (transformed into positional encoding).

The total number of atoms is also learned with a special penalty which is added to the NeRF recunstruction loss

The method is tested on two synthetic and one real datasets, and present competitive resutls
The authors provide an ablation study to justify the two main modules, namely the dictionary size penalty and the perturbation MLP

**Summary Of The Review:**

This seems like a promising method. If my concerns are addressed, I see no reason to accept this paper.

---

The authors have addressed my concerns, and I think this paper is clearer now.

---

> ### Author Response · Authors · 2022-11-12
> **Author Response**
>
> Thank you for your thorough review. We have modified the paper to reflect your feedback and are interested in any further feedback or questions you may have. We have summarized the changes made to the new version of the paper in top-level comment titled “Summarization of Changes”. Below you will find our responses to your question and comments.
>
> **It seems that the input, as in PixelNeRF, images are needed in inference time, otherwise - where is the feature map obtained from to query the dictionary? If this is so - this is a major caveat compared to other NeRF methods that can render an image using only the MLP.**
>
> You’re correct that conditional NeRF methods such as PixelNeRF, uORF, and NRC require images at inference whereas single-scene NeRF methods only require images during training. We do not consider this to be a drawback as the use-cases of conditional NeRF and single-scene NeRF methods are different. Single-scene NeRF methods only render views of the one scene they were trained for whereas conditional NeRF methods render views of new scenes. Further, the goal of NRC and other object-centric work is to learn to decompose new scenes into their semantic parts.
>
> **Another possible discrepancy is the fact that the atom is generated from 2D coordinates of a view different from the novel one - but unlike "PixelNeRF", if an object is occluded in the "input view" we will get a wrong atom and hence the wrong class for the segmentation map - is this resolved by a low density for the rendering process as well?**
>
> We appreciate the feedback and have revised the section *Image Encoding and Camera Projection* to provide better clarity on this point. During the training process the camera projection of the feature map is performed before the atom/codebook assignment. To be clear, the MLP is conditioned on the predicted segmentation map from the novel view. To produce segmentations of the input view during inference we set the camera projection to be identity.
>
> **How exactly is a new atom added to the dictionary? Specifically, when the parameter 's' "signifies when to add a new code" - what value is the atom given?**
>
> We initialize new codes with a standard normal distribution. During training we have k+1 codes where k is the current number of learned codes. The k+1 code is needed by the straight-through-estimator to optimize for s. If using the k+1 code decreases reconstruction loss, then the value of s is increased by back-propagation. We have added these details to the *Assigning + Learning Codes* subsection of methods.
>
> **I would recommend using a different notation for the two input images, rather than h and h'.**
> Thank you for the feedback. We have made changes to reflect your recommendation and changed images notation to $I_h$ and $I’_h$.
>
> **Figure 2 - It is not clear which part of the pipeline that are included in the MLP training, are not included in inference. For example: Where is the feature map F generated from? What feature values (or atoms) will we get for queries of regions out of the "input view"?**
>
> We have revised the first two paragraphs of the methods section and edited figure 2 to improve clarity about the training pipeline. Additionally we’ve added a section “Using the Codebook for Downstream Tasks” to clarify how NRC is used at inference based on your feedback.
>
> To address your specific questions:
> - The feature map is generated from processing the image through the convolutional network (encoder).
> - For queries completely outside of the input view (i.e. the novel view is rotated 180 degrees from the input), the feature map will be projected to the novel view, but the reconstruction will be poor. We discuss this problem of finding appropriate corresponding views which current 3D object-centric works have in the limitations section.
>
>
> **There are some partially similar works, but this is a field with a very high publication rate, so this is not a reason to reject.**
>
> Thank you for pointing us to these papers. We have incorporated them into our related works.

---

> > ### Comment · Reviewer_77Ao · 2022-11-18
> > **Leaning towards accept**
> >
> > I thank the authors for the detailed response and appreciate the work done revising the manuscript.
> > The modifications made make the method easier to understand, IMO.
> > My concerns have been addressed, at large.

---

> > > ### Author Response · Authors · 2022-11-19
> > > **Thank you**
> > >
> > > We thank the reviewer for their response, appreciation of the rebuttal and increase of the score.
> > >
> > > We are happy to have further discussions if you have any other concerns.

---

### Official Review · Reviewer_PdhP · 2022-10-27

**Confidence:** 3
**Correctness:** 4
**Technical Novelty And Significance:** 3
**Empirical Novelty And Significance:** 3
**Recommendation:** 6

**Clarity, Quality, Novelty And Reproducibility:**

This paper is clear and easy to understand. It might be better if the authors could summarize the contribution of this method so that readers and easily get the key information.
The method is simple and clearly described. It should not be too hard to implement it. The authors did not mention whether the code will be open source.
The idea of using a codebook to represent a scene is not novel, but using volume rendering and NeRF to learn the object-level codebook is novel.

**Details Of Ethics Concerns:**

No ethics concerns.

**Strength And Weaknesses:**

Strength:
- Using an object-level codebook to represent a scene is not a new idea (MarioNette). But this paper makes the learned representation more effective with various techniques.
- The experiment is through by comparing many related methods on three distinct tasks. The authors also show the method is not only useful for synthetic data but also for more complex real data.

Weakness:
- The majority of the experiment is done using synthetic datasets like THOR. The scene is relatively simple and has less clutter and occlusion than typical real-world scenarios. It is unclear how much the degradation will be for a more complex scenario.

**Summary Of The Paper:**

This paper decouples the scene representation into a set of learned codes. The codebook is object-level. The feature volume of a scene is encoded using the codebook before rendering with an MLP (NeRF). Experiments show the effectiveness of the proposed representation on various downstream tasks. The representation is evaluated using three tasks, object navigation in THOR, unsupervised segmentation in various datasets, and depth ordering estimation. These tasks demonstrate the effectiveness of the learned codebook.

**Summary Of The Review:**

This paper presents an effective way to learn object-level codebooks through volume rendering. It shows the effectiveness of this codebook with various experiments. It is unclear whether this method is robust to cluttered scenarios, with more occlusions. I think overall it is a nice paper and should be valuable for the community to read.

---

> ### Author Response · Authors · 2022-11-12
> **Author Response**
>
> Thank you for your thorough review. We have modified the paper to reflect your feedback and are interested in any further feedback you may have. We have summarized the changes made to the new version of the paper in top-level comment titled “Summarization of Changes”. Below you will find our responses to your question and comments.
>
> **The majority of the experiment is done using synthetic datasets like THOR. The scene is relatively simple and has less clutter and occlusion than typical real-world scenarios. It is unclear how much the degradation will be for a more complex scenario.**
>
> Thank you for the suggestion. We have added examples of cluttered scenes from NYU Depth to appendix A.1 (figure 4) in the revised version. As you’ve mentioned, the segmentation performance degrades for NYU Depth compared to THOR and CLEVR (Table 1). We’d like to point out that NRC handles clutter and occlusion better than existing object-centric methods (29% relative improvement on NYU Depth). In particular, NRC handles clutter better than uORF and ObSuRF because the rendering and memory cost does not scale with the number of objects in the scene. In appendix A, we report unsupervised segmentation performance on NYU Depth for scenes with 5 or more objects and find 38% relative improvement over existing methods.
>
> In addition, we trained NRC on Matterport-3D scenes based on the feedback from reviewer GeqQ which are visually more realistic. Qualitative and quantitative segmentation results have been added to appendix A (figure 6, table 6) and can be found in the “MatterPort-3D Experiments” section. We are currently running point-navigation experiments for Matterport-3D and will update the revision. **Update -** We have run the point navigation experiments in Habitat. We find similar conclusions as object-navigation in THOR. NRC is capable of handling more complex and visually realistic environments compared to other 3D object-centric methods. We've added qualitative examples to the supplemental in the *Object Navigation/gibson* folder and quantitative results to the appendix (table 7).
>
> **This paper is clear and easy to understand. It might be better if the authors could summarize the contribution of this method so that readers can easily get the key information.**
>
> Thank you for the suggestion. We’ve edited the last two paragraphs in the introduction to better clarify our contributions. To briefly summarize here, our contributions are the following:
>
> - We propose Neural Radiance Field Codebooks (NRC). NRC learns a codebook of object categories which are composed to explain the appearance of 3D scenes from multiple views. This learned representation is effective for segmentation as well as geometry-based tasks such as object navigation and depth ordering.
>
> - NRC resolves current limitations in existing 3D object-centric methods. First, NRC's method for assigning object codes to regions of the image enables constant rendering cost whereas other methods scale with the number of objects. Second, we introduce a mechanism for differentiably adding new categories which allows the codebook to scale with the complexity of the data. Finally, modeling intra-category variation in conjunction with the codebook allows semantic and geometric object information to be shared across scenes.
>
> - We empirically show the utility of NRC representations for unsupervised segmentation, object navigation, and depth order. To our knowledge, our results on NYU Depth are the first for unsupervised segmentation on complex, real-world scenes. We show 29% relative improvement compared to object-centric methods on the NYU Depth dataset. For object navigation and depth ordering we observe 3.1% improvement in navigation success rate 5.5% improvement in depth ordering accuracy.
>
> **The authors did not mention whether the code will be open source.**
>
> Code will be open-sourced and we’ve added implementation details in Appendix B.1. We also included source code in the supplementary.

---

### Author Response · Authors · 2022-11-12
**Summary of Changes**

We would like to thank the reviewers for their thorough feedback. In response to their valuable suggestions we have made significant revisions. We summarize the changes by section:

### Introduction
We have incorporated the suggestion from reviewer PdhP to more clearly summarize our contributions. This is reflected in the last two paragraphs of the introduction.


### Related Works
We appreciate reviewer 77Ao pointing us to similar papers and have incorporated them into our related works.

### Method
Based on feedback from reviewers we have made significant revisions to the method section to improve clarity.
- Based on the suggestion from reviewer 77Ao and BHLx we have changed the subsection “Assigning Categorical Codes” to “Assigning and Learning Categorical Codes”, and added further details about how the codebook is trained.
- Based on suggestions from reviewer 77Ao and BHLx we have added a subsection to methods called “NRC For Downstream Tasks”, which details how the various components of NRC are used for segmentation, depth ordering, and object navigation. We have also incorporated suggestions from all reviewers on changes to notation and improving the method diagram (figure 2).

### Experiments
Reviewer PdhP had questions about how well NRC would perform on more cluttered and complex real-world scenes. We have provided further qualitative examples of cluttered NYU Depth scenes to appendix A (figure 4) and quantitatively compared NRC on scenes from NYU Depth with 5 or more objects (Table 5). We find that NRC can scale to a larger number of items per scene, although it struggles on objects that differ significantly from the training distribution.

Reviewer GeqQ has raised concerns about application of NRC to real-world data. We have incorporated the feedback and trained NRC on scenes from Matterport3D. Quantitative and qualitative results for unsupervised segmentation have been added to appendix A (table 6 and figure 6). Additionally we are running point navigation experiments on Matterport3D.

**Update** -  We have run Habitat Point Navigation experiments to compare NRC with other visual encoders. Qualitative examples can be found in the *Object Navigation/gibson* folder of the supplemental. Quantitative comparison can be found in table 7 of the appendix.

We would also like reviewers to note that object-centric learning for real-world data is an open challenge, and to our knowledge we are the first to show unsupervised segmentation results on complex real-world scenes such as NYU Depth.

Reviewer BHLx recommended adding qualitative examples for depth order, object navigation, and view synthesis with further analysis. We added qualitative examples of each to appendix A (see “object navigation” and “3D segmentation” videos in the supplemental and figure 5-8). We have also added further analysis in the results subsection of experiments.

---

### Public Comment · ~Jonathan_C_Balloch1 · 2023-05-03
**Great potential to expand on this work given open source code**

Authors, congratulations on the acceptance of your paper; this work is greatly impactful, especially considering the effective use of an *expanding* codebook, as opposed to the static discretization used by prior models. Do you already have a code implementation, or plans to open source the code?

---

> ### Author Response · Authors · 2023-05-03
> **Thanks for the appreciation, link to code**
>
> Thanks for the kind words. We are presenting this work in ICLR this week.
>
> Here is the code: https://github.com/MattWallingford/NeuralRadianceFieldCodebooks
>
> Let us know if you need anything else.

---

### Decision · Program_Chairs · 2023-01-20

**Decision:**

Accept: poster

**Justification For Why Not Higher Score:**

This is interesting approach but still preliminary

**Justification For Why Not Lower Score:**

This interesting approach is promising and worth sharing with ICLR researchers

**Metareview: Summary, Strengths And Weaknesses:**

The paper aims to improve Neural Radiance Field with the concept of codebook for object categories to deal with the appearance of 3D scenes from multiple views, by allowing semantic and geometric object information to be shared across scenes. The idea of using a codebook to represent a scene is not novel (e.g. MarioNette), but adopting this for NeRF to enable the discovery of reoccurring visual and geometric patterns across scenes is interesting. Experiments cover both synthetic and real images with good results.


**Note From Pc:**

if the above contains the word "oral" or "spotlight" please see: "oral" presentation means -> notable-top-5% and "spotlight" means -> notable-top-25%. As stated in our emails, we are disassociating presentation type from AC recommendations

**Summary Of Ac-Reviewer Meeting:**

although 2 reviewer gave 5 (besides 8 and 6 by the other 2 reviewers), the approach is interesting and promising, and the authors have revised the paper significantly in response to the reviews.